# Evaluating the efficacy of Internet-Based Exercise programme Aimed at Treating knee Osteoarthritis (iBEAT-OA) in the community: a study protocol for a randomised controlled trial

Sameer Akram Gohir,[1] Paul Greenhaff,[1,2,3,4,5] Abhishek Abhishek,[1,6] Ana M. Valdes[1]

For numbered affiliations see end of article.

**Correspondence to**
Sameer Akram Gohir;
sameer.gohir@nottingham.ac.uk

## ABSTRACT

**Introduction** Knee osteoarthritis (OA) is the most common joint disease worldwide. As of today, there are no disease-modifying drugs, but there is evidence that muscle strengthening exercises can substantially reduce pain and improve function in this disorder, and one very well tested physiotherapy protocol is the 'Better Management of Patients with Osteoarthritis' developed in Sweden. Given the high prevalence of knee OA, a potentially cost-effective, digitally delivered approach to treat knee OA should be trialled. This study aims to explore the benefits of iBEAT-OA (Internet-Based Exercise programme Aimed at Treating knee Osteoarthritis) in modulating pain, function and other health-related outcomes in individuals with knee OA.

**Methods and analysis** A randomised controlled trial was designed to evaluate the efficacy of a web-based exercise programme in a population with knee OA compared with standard community care provided by general practitioners (GPs) in the UK. We anticipate recruiting participants into equal groups. The intervention group (n=67) will exercise for 20–30 min daily for six consecutive weeks, whereas the control group (n=67) will follow GP-recommended routine care. The participants will be assessed using a Numerical Rating Scale, the Western Ontario and McMaster Universities Osteoarthritis Index, the Arthritis Research UK Musculoskeletal Health Questionnaire, the Pittsburgh Sleep Quality Index, 30 s sit to stand test, timed up and go test, quantitative sensory testing, musculoskeletal ultrasound scan, muscle thickness assessment of the vastus lateralis, and quadriceps muscles force generation during an isokinetic maximum voluntary contraction (MVC). Samples of urine, blood, faeces and synovial fluid will be collected to establish biomarkers associated with changes in pain and sleep patterns in individuals affected with knee OA. Standard parametric regression methods will be used for statistical analysis.

**Ethics and dissemination** Ethical approval was obtained from the Research Ethics Committee (ref: 18/EM/0154) and the Health Research Authority (protocol no: 18021). The study was registered in June 2018. The results of the trial will be submitted for publication in a peer-reviewed journal.

**Trial registration number** NCT03545048

## Strengths and limitations of this study

► This study is the first to evaluate a web-based exercise intervention to improve pain in sufferers of knee osteoarthritis (OA) in the UK.

► We plan to recruit only those individuals presenting with radiographic evidence of knee OA that will rule out non-OA causes of knee pain.

► This study is the first to evaluate changes in sleep disturbance due to knee OA along with an exercise intervention.

► The lack of double-blind study design is typically a limitation of exercise and lifestyle intervention studies.

► Six weeks of iBEAT-OA (Internet-Based Exercise programme Aimed at Treating knee Osteoarthritis) intervention may not show a statistically detectable reduction in knee pain and may be a potential limitation of this study.

## INTRODUCTION

Knee osteoarthritis (OA) is a common cause of disability globally and is mostly managed in primary care.[1] In the UK, 10% of people between the ages of 65 and 74 consult their general practitioners (GPs) for OA every year.[2] In the UK, 1 in 25 people consult their GP for knee OA annually.[1]

The first line of treatment for OA involves exercise, education and weight loss if applicable.[3–7] Exercise improves joint and patient-centred outcomes in knee OA, and encourages them to manage activities of daily living.[8–21] In fact, strengthening knee exercises have been shown to reduce progression of radiographic changes of knee OA.[22] A recent Cochrane review on exercise interventions for knee OA has reported that exercise significantly reduced pain (12 points/100; 95% CI 10 to 15) and improved physical function (10 points/100; 95% CI 8 to 13)

to a moderate degree.[23] Another systematic review and meta-analysis conducted on the effect of resistance exercises in patients with knee OA reported that resistance training relieved pain (standard mean difference: −0.43; 95% CI −0.57 to −0.29).[24] There is a disparity of opinion regarding the effectiveness of different types of exercise for knee OA to reduce pain, and a combination of open and closed isotonic exercises is recommended for knee OA.[25] Symptoms of knee arthritis can be exacerbated by following ineffective or unsafe exercises regimens,[26–29] leading to poor prognosis and poor adherence to exercise intervention,[30] hence the need to choose the type and frequency of exercise intervention carefully.

Versus Arthritis recommends knee exercise for 'knee pain'; however, these exercises are not specific to knee OA and include soft tissue injuries of knee as a cause of knee pain.[31] An exercise regimen is normally recommended by GPs as part of the first line of treatment when they consult someone with arthritis-related knee pain. If an exercise intervention fails to impact positively on pain perception, patients are referred to physiotherapy. After a comprehensive assessment by physiotherapists, a variety of exercise interventions that fluctuate in exercise modality, intensity and duration can be prescribed, but with variable success in terms of outcome. Therefore, there is a need for standardised exercise interventions to address knee OA pain, which would ideally be accessible online to maximise efficacy and cost-effectiveness. iBEAT-OA (Internet-Based Exercise programme Aimed at Treating knee Osteoarthritis) is the first randomised controlled trial (RCT) conducted on the app-based knee exercises developed by Joint Academy (JA). The exercise intervention (JA App), which we will use in this project, is a web-based programme derived from the Supported Osteoarthritis Self-Management Programme (SOASP) also known as the 'Better Management of Patients with Osteoarthritis' (BOA) developed in Sweden.[32] Essentially, the JA App is a digital version of face-to-face BOA. From January 2008 until January 2017, around 75 000 patients participated in the SOASP, 2339 physiotherapists and occupational therapists were educated to deliver the SOASP, and today SOASP is offered in 700 clinics all over Sweden.[32] This intervention has been rated good to very good by 94% of patients at reducing pain related to knee OA and has an excellent safety profile and acceptability.[33] A quasi-experimental study done on the internet-based JA programme has shown a change in mean Numerical Rating Scale (NRS) score which was larger than the minimal clinical difference (5.4 vs 4.1; p<0.001).[34]

There exist a number of other exercise programmes within care programmes for people with knee OA pain. Among these is the Good Life with osteoArthritis in Denmark (GLA:D) programme, which is a registry-based study that implemented clinical guidelines (patient education and exercise but not weight loss) to patients with knee OA.[35] The programme consists of a 2-day training course for physiotherapists, including training to diagnose and deliver OA care, and an 8-week supervised exercise intervention for patients with OA, with a minimum intervention of three sessions in total. However, GLA:D does not deliver intervention via a web or smartphone app.

The ESCAPE (Enabling Self-management and Coping with Arthritic Pain using Exercise) app is a support tool for people who have already attended a person-to-person ESCAPE programme[18] and enables participants to continue to exercise safely in the home environment following the person-to-person programme. A limitation of this approach is that the app is a support tool for the main programme and provides reminders for exercise. It cannot be used as the sole basis for treatment and care of knee OA. The recommendation is to use it in conjunction with the advice and professional judgement of the GP or other healthcare practitioners, which is the limitation of this app.

Because of the high prevalence of knee OA, strategies to deliver exercise interventions that are both efficacious and inexpensive should be prioritised. Some of the issues relating to the delivery of exercise interventions for knee OA include compliance, accessibility to clinics for people with mobility problems and the cost of delivering such services. There are previous studies which have assessed the efficacy of home-based exercises[10 12 36]; however, very few studies looked at web-based delivery of exercise intervention.[34 37–42] Most of these studies recruited patients with knee pain, and only two studies assessed for radiographic evidence of knee OA.[39 41] This makes the results of these studies less specific to base a recommended treatment for knee OA. In addition, none of these studies has investigated the role of exercise interventions for pain relief, assessing levels of pain sensitivity and central sensitisation, which are likely different for knee OA and other causes (eg, soft tissue injuries) of knee pain.

The most recent RCT which studied the benefit of internet-based exercises versus routine physiotherapy reported no difference on the Western Ontario and McMaster Universities Osteoarthritis Index (WOMAC) at 12 months between internet-based versus face-to-face physiotherapy, and suggested further studies with strategies to maximise the benefit of exercises-based interventions for patients with knee OA.[41] Therefore, there is a need for specific, efficacious and cost-effective exercise programme that individuals can perform in the home setting.

Knee OA pain is accompanied by a number of additional disturbances that influence the individual's health, which we also propose to study. There is a well-known link between sleep disturbance and chronic pain, and epidemiological studies[43–45] and experimental studies[46–50] have established a link between disturbed sleep and knee OA. These studies suggest that individuals with OA have greater sleep disturbances. Furthermore, sleep disturbance is recognised as an important factor in determining pain perception.[50] A relationship between sleep disturbance and pain severity in patients with knee OA is usually explored, and sleep disturbances such as shortened sleep duration and fragmented sleep[43] have been

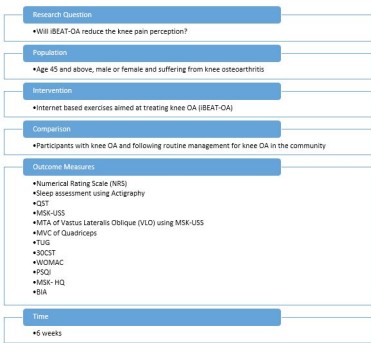

**Figure 1** A randomised controlled trial evaluating the efficacy of Internet-Based Exercise programme Aimed at Treating knee Osteoarthritis (iBEAT-OA)—PICOT format. 30CST, 30 s sit to stand test; BIA, bioimpedance analysis; MSK-HQ, Arthritis Research UK Musculoskeletal Health Questionnaire; MSK-USS, musculoskeletal ultrasound; MTA, muscle thickness assessment; MVC, maximum voluntary contraction—isokinetic contraction of the quadriceps muscle; OA, osteoarthritis; PSQI, Pittsburgh Sleep Quality Index; QST, quantitative sensory testing, TUG, timed up and go test; WOMAC, Western Ontario and McMaster Universities Osteoarthritis Index.

associated with increased sensitivity to pain in patients with OA, and consequently with decreased quality of life.[45] Therefore, studying the sleeping pattern is highly relevant if we are discussing a successful online exercise programme for knee OA.

Recent epidemiological and clinical studies have also underlined that metabolic syndrome has the most significant impact on the initiation and severity of OA.[51–55] Metabolic-triggered inflammation, also known as meta-inflammation,[56] can be a result of abnormalities in body composition, adipokines, cytokines, lipids and vitamin D and has been associated with the pathogenesis of OA.[54] Given that body composition is influenced by a number of the variables associated with OA, for example, systemic inflammation, sleep patterns, altered physical activity levels and altered energy levels, it is important to study this as well. The intricate links between exercises, sleep, pain, metabolic syndrome, body composition and OA are not fully understood. To date, the relationship between improvements in pain due to exercise for pain relief and other changes related to health parameters that have been linked to OA or chronic pain has not been explored. Specifically, the effects of exercise for knee pain relief on sleep, biomarkers of inflammation and insulin resistance are unknown. We aim to study these parameters and establish the link between exercises for knee OA and these parameters. Figure 1 shows this in PICOT format.

The efficacy of web-based delivery of exercises has not been evaluated in a UK setting. This is the first study to conduct web-based exercises intervention in the UK. This study is also novel because it will be the first in the UK to bring together a wide range of factors influencing knee OA. This will include quantitative sensory testing (QST), which is deemed to be an objective measure, along with

measurement of sleep patterns undertaken using an activity monitor (actigraphy).

### Rationale
This study aims to explore the benefits of an internet-based exercise programme in patients with knee OA to establish if 6-week intervention reduces pain perception and sensitivity. Being a web-based exercise intervention also makes it accessible and cost-effective to volunteers and physiotherapists, and will hopefully help establish a standardised intervention for knee OA that will maintain individual motivation, compliance and managing pain. The study will also endeavour to explore the complicated relationship between chronic pain, sleep, biomarkers of inflammation, body composition and knee OA. Therefore, this study could potentially help generate recommendations for the treatment of knee OA.

### Primary objective
The primary objective is to test whether an internet-based exercise intervention can reduce pain perception (NRS) in knee OA.

### Secondary objective
The secondary objective is to test whether there is a benefit of iBEAT-OA in improving sleep disturbances and in reducing pain sensitisation and metabolic syndrome.

### METHODS AND ANALYSIS
### iBEAT-OA trial design
The iBEAT-OA study is an RCT in the primary care setting in Nottingham with participants identified as having knee OA, randomised 1:1 to web-based exercises or usual care, as shown in figure 2.

### Setting
The study will be in a home-based, primary care setting.

### Patient and public involvement
At the patient and public involvement (PPI) representative meeting on 14 December 2015, seven volunteers suffering from chronic OA pain were asked about the relevance of studying sleep in the context of OA pain. They all were very supportive of studying and understanding sleep patterns. They viewed actigraphy as a good non-invasive alternative to polysomnography and saw no problem with using the device for 6 weeks.

On 17 May 2017, six representatives from the PPI musculoskeletal group were asked about exercise interventions and were all supportive of this. They were also asked about the extraction of synovial fluid from their joints. Five out of six said they would not have a problem with this if it was performed by a specialist using ultrasound to guide the needle. This approach was therefore adopted as an optional test within the study.

The latest version of the patient facing documentation (participant information sheet, consent form, invitation

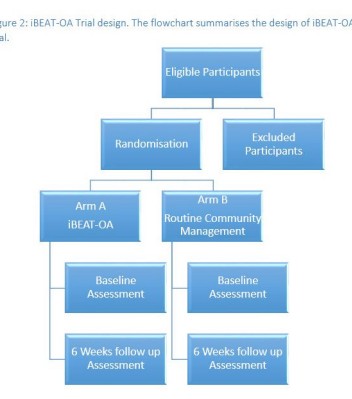

Figure 2: iBEAT-OA Trial design. The flowchart summarises the design of iBEAT-OA trial.

**Figure 2** iBEAT-OA trial design. The flow chart summarises the design of iBEAT-OA trial. iBEAT-OA, Internet-Based Exercise programme Aimed at Treating knee Osteoarthritis.

letter and recruitment flyer) have been shared with the PPI group and received favourable comments.

### Recruitment

A selection of eligible people for the study will be invited from existing databases[57] held at Academic Rheumatology, City Hospital Nottingham of participants with knee pain who have agreed to be contacted for future studies. The inclusion and exclusion criteria are shown in figure 3. Any shortfall in the recruitment will be compensated by sending study leaflets to GP surgeries. The GPs will follow the inclusion and exclusion criteria. All those individuals who are suitable for the study will be sent the study information sheet. Those who return the completed screening consent form will be contacted and screened for inclusion in the study.

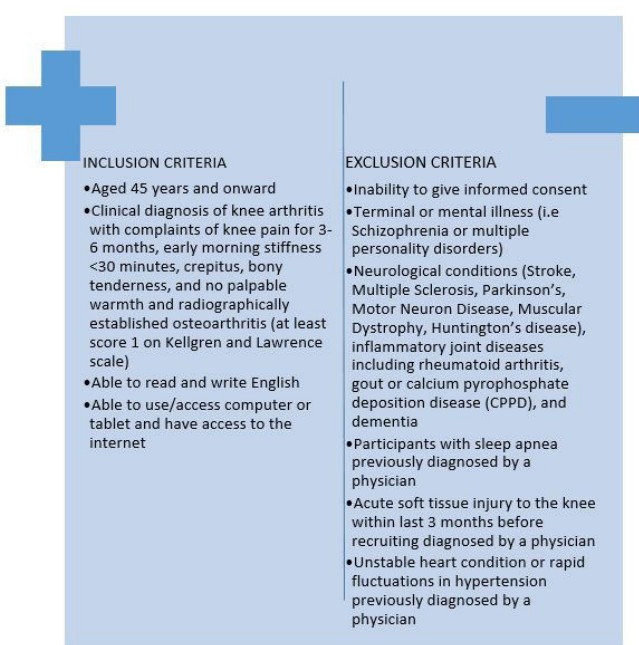

**Figure 3** Inclusion and exclusion criteria for iBEAT-OA trial. iBEAT-OA, Internet-Based Exercise programme Aimed at Treating knee Osteoarthritis.

### Randomisation and blinding

Eligible participants will undergo computer-generated randomisation and allocation concealment. Randomisation will be done using 'sealed envelope' (https://www.sealedenvelope.com/), an online randomisation service. Participants will be equally allocated between treatment arms with at least n=67 per arm and this will be done by the research team. At this point, individuals will be alerted to which study group they have been assigned, and hence blinding is not possible. Because this might affect the motivation of participants randomised into the control arm, these individuals will be offered access to the web-based exercise programme after completing the study. Blinding of exercise intervention studies is difficult; however, the clinical research team will be blinded to the intervention.

### Sample size and justification

A Swedish study using specifically JA as the intervention and NRS as the outcome reported an effect size of 1.3 on the NRS,[34] which corresponds to 0.56 SD (average SD 2.3). A study with an expected 60 participants per arm at the end of the 6-week intervention has 86% power to achieve such an effect size with an alpha of 0.05. However, a recent systematic review of 44 high-quality exercise trials for knee OA pain (3537 participants)[23] found an average effect of 12/100 Visual Analogue Scale (VAS) points corresponding to 0.49 SD. A sample size of n=60 per group is necessary to achieve 75% statistical power. The estimated dropout rate for exercise interventions is 10%; therefore, a sample of 67 per arm will be recruited to achieve between 75% and 86% power, that is, 80% power with an alpha level of 0.05.

### Intervention

iBEAT-OA will use a web-based exercise platform known as JA, as recent pilot studies[34 40] demonstrated promising results. This programme is based on a Swedish face-to-face self-management programme known as 'Artrosskolan' (*The Osteoarthritis School*), which provided structured information and exercises for knee OA to a relevant population suffering from knee OA. This was initially introduced as 'Better Management of Patients with Osteoarthritis'.[58] Compliance to this self-management programme has been reported as good. Of 20 200 patients, 62% reported daily use of this programme at 3-month follow-up; however, this percentage drops to 37% at 12-month review.[33] The company that produced the JA platform has given permission to use their web-based app to conduct this study.

The exercise intervention comprises a mixture of open and closed chain exercise manoeuvres, and a combination of concentric and eccentric exercise modalities, and is focused on the overall strength of legs including the muscles around the hips and knee joints. An open kinetic chain is defined as 'a combination of successively arranged joints in which the terminal segments can move freely'.[59] Closed chain exercises are exercises or movements where the distal aspect of the extremity is fixed to an object that is stationary and proximal joints move.[59] There are

balance enhancement exercises as well. There are educational sessions integrated into the programme covering the basics of OA, its treatment, self-managing symptoms of OA and the benefits of maintaining a healthy lifestyle.

## Control group

The control group will continue with routine self-management or GP management of knee OA which is offered in the community setting in the UK.

## Follow-up duration

Duration of follow-up will be 6 weeks postintervention.

The 6-week duration has been selected for this study based on a previous observational quasi-experimental study which reported 6 weeks of web-based treatment were effective to reduce knee pain.[34]

## Research assessment

Outcome Measures in Rheumatology-Osteoarthritis Research Society International has recommended six domains as mandatory to be measured and reported in all hip and knee OA clinical trials.[60] These are pain, physical function, quality of life, patient's global assessment of the target joint, adverse events including mortality, and joint structure (in specific circumstances and depending on the intervention). The first four domains are measured by NRS, QST, WOMAC and Arthritis Research UK Musculoskeletal Health Questionnaire (MSK-HQ). The adverse events will be reported at the end of the trial, and joint structure will be reported based on X-rays and the findings of ultrasound scan of the most painful knee. Time up and go test and 30 s sit to stand test (30CST) are objective assessments of physical functions based on the recommendations of Osteoarthritis Research Society International[61]; thus, the following outcome measures will be assessed:

► Sleep.
► NRS (average pain on the day of assessment on a scale of 0–10).
► QST (pressure pain threshold, temporal summation and conditional pain modulation).
► Inflammatory markers on ultrasound (synovial fluid, synovial hypertrophy and hypervascularity), using musculoskeletal ultrasound scan (MSK-USS).
► Muscle thickness assessment (MTA) of vastus lateralis oblique using MSK-USS.
► Maximum voluntary contraction (MVC)—isokinetic contraction of the quadriceps muscle.
► Biomarkers of insulin resistance.
► Physical functioning (time up and go test, 30CST).
► WOMAC.
► Pittsburgh Sleep Quality Index (PSQI).
► General health questionnaire (MSK-HQ).
► Body composition assessment by bioimpedance analysis.

## Start and end dates

The trial has started in February 2019 and enrolment will end in December 2020.

## Description of intervention

Participants seeking care for OA will be informed of the study both orally and in writing. Those who qualify for the study will give signed informed consent, and the participants will be randomised to an intervention or control group. The consent will be taken by an experienced member of the research team. The exception applies to those participants who have not had knee X-rays in the previous 12 months. These individuals will be called to the hospital, and after gaining valid consent knee radiographs will be obtained and their X-rays will be assessed by experienced staff. Once a definite eligibility criterion (Kellgren and Lawrence (K/L) score at least 1 or above) is established in such cases, the qualifying participants will be randomised to the intervention or control group.

The intervention group will have an assessment session with experienced staff, and NRS,[62–64] WOMAC,[65 66] MSK-HQ,[67 68] PSQI,[69–72] 30CST,[73–75] timed up and go test (TUG),[76 77] QST,[78–84] MSK-USS,[85–89] MTA of vastus lateralis,[90 91] quadriceps muscle force generation during an isokinetic MVC,[92] and urine and blood samples[93 94] will be taken at baseline (refer to online supplementary file for further information). These samples will be assessed for circulating levels of fasting insulin, glucose, C reactive protein, triglycerides, low-density lipoprotein, high-density lipoprotein, tumour necrosis factor-alpha, interleukin-6 and interleukin-1. Those who consent for aspiration of synovial fluid will go through an ultrasound-guided aspiration procedure.[85–88] The intervention group will be given an actigraphy device (ActTrust). Their sleeping pattern will be recorded quantitatively.

The intervention group will then receive a link via email, which will be used to log in to the JA online portal (see https://www.jointacademy.com). After log-in has been achieved, the exercise intervention will start. This will consist of a 6-week internet-based physical therapy programme that will provide information, exercise, contact details of a personal physiotherapist, education about lifestyle and behavioural changes. This intervention can be accessed using a smartphone or a tablet . The programme also encourages physical activity adherence by sending email prompts on a regular basis. Initially, participants will answer an online questionnaire covering areas such as joint pain intensity, health-related quality of life, physical function, as well as performing a physical test assessing lower limb strength. This questionnaire will form part of the online baseline assessment. The exercise intervention will consist of knee and hip exercises along with some functional activities such as sit to stand and stairs climbing.

After the 6-week exercise intervention, participants will fill in the same questionnaire and perform the same physical tests to enable evaluation. There are two face-to-face meetings between participants and physiotherapist/nurse, at enrolment and after 6 weeks. The physiotherapist will be available via asynchronous online chat or over the phone during the 6-week study period.

The control group will continue with their routine self-management which is offered in the community set-up. They will be assessed on NRS, QST, WOMAC, MSK-HQ, PSQI, 30CST, TUG, MSK-USS, MTA of vastus lateralis, MVC of the quadriceps muscle, body composition, and urine and blood samples at baseline. The control group will also use actigraphy to monitor sleep patterns. They will follow the routine management of knee OA recommended by the National Institute for Health and Care Excellence (NICE), which includes non-pharmacological and pharmacological management.[95] They will be reassessed after 6 weeks on the same outcome measures to determine if there has been an impact of self-management strategies.

### Data management

In the iBEAT-OA trial, data will be collected during the first and last session. Additional, weekly pain scores will be collected via an online portal, and actigraphy will be used to collect sleep patterns. Online portal and actigraphy data will be collated by trained local research staff, and data entry in a relational Microsoft Access database will be completed in a standardised fashion. The clinical research forms from the first and last study visit will be sent to the data entry site. A central data manager performs and monitors data entry. This will include questionnaires (MSK-HQ, PSQI, WOMAC) and data from QST, 30CST, TUG, MSK-USS, MTA of vastus lateralis, body composition and MVC of the quadriceps muscle.

### Statistical analysis

Clinical trial data will be analysed using an intention-to-treat approach. We will compare outcome measures (eg, pain sensitivity, pain scores, sleep patterns and inflammatory measures) between the exercise and non-exercise groups controlling for baseline scores using appropriate parametric and non-parametric statistical tests. Additional observational secondary analyses (ie, correlations between change in sleep patterns and change in pain measures) will be carried out using parametric statistics and adjusting for the relevant covariates. These missing data will be calculated based on multiple imputation. The SPSS package will be used for statistical analyses.

### Adverse events

There are no serious adverse events reported with these exercises. These exercises have been trialled on 75 000 patients from 2008 to 2017 with no serious adverse events.[33 58] There is a small chance of an increase in the knee, hip or back pain.[23] We will monitor the pain levels of the patient on a weekly basis using an internet-based interface to monitor any increase in the knee, hip or low back pain. If the pain exacerbates to the level that the participant starts struggling with the activities of daily living, then they will be advised to stop participating in the study and to contact their GP.

### Criteria for terminating the study

As the study involves only two assessments and does not involve investigational medicinal products or medical devices, and the same intervention has not given rise to any serious adverse events in over 70 000 participants in Sweden, it is not envisaged that circumstances will arise that require termination of this study.

### Ethics and dissemination

The study results will be submitted to Versus Arthritis, regulatory authorities and a peer-reviewed journal for publication. Also, the results will be presented at national and international conferences. Study participants will also be informed of the results if requested.

### Insurance and indemnity

Insurance and indemnity for clinical study participants and study staff are covered within the National Health Service (NHS) Indemnity Arrangements for clinical negligence claims in the NHS, issued under the cover of Health Service Guidelines HSG (96)48. There are no special compensation arrangements, but study participants may have recourse through the NHS complaints procedures.

The University of Nottingham as research sponsor indemnifies its staff, research participants and research protocols with both public liability insurance and clinical trials insurance. These policies include provision for indemnity in the event of a successful litigious claim for proven non-negligent harm.

### Study conduct

Study conduct will be subject to systems audit for inclusion of essential documents; permissions to conduct the study; Curriculum Vitaes (CVs) of study staff and training received; local document control procedures; consent procedures and recruitment logs; adherence to procedures defined in the protocol (eg, inclusion/exclusion criteria, timeliness of visits); and accountability of study materials and equipment calibration logs.

### Study data and audits

Monitoring of study data shall include confirmation of informed consent; source data verification; data storage and data transfer procedures; local quality control checks and procedures; back-up and disaster recovery of any local databases; and validation of data manipulation. This will be managed by the direct study team.

Entries on study forms will be verified by inspection against the source data. A sample of the forms (10%) will be checked on a regular basis for verification of all entries made. In addition, the subsequent capture of the data on the study database will be checked. Where corrections are required, these will carry a full audit trail and justification.

Study data and evidence of monitoring and systems audits will be made available for inspection by the Research Ethics Committee (REC) as required.

## Record retention and archiving

In compliance with theInternational Council for Harmonisation of Technical Requirements for Pharmaceuticals for Human Use (ICH) and Good Clinical Practice (GCP) guidelines, regulations and in accordance with the University of Nottingham Code of Research Conduct and Research Ethics, the chief or local principal investigator will maintain all records and documents regarding the conduct of the study. These will be retained for at least 7 years or for longer if required. If the responsible investigator is no longer able to maintain the study records, a second person will be nominated to take over this responsibility.

The study master file held by the chief investigator on behalf of the sponsor shall be finally archived at secure archive facilities at the University of Nottingham. This archive shall include all anonymised study databases and associated meta-data encryption codes.

## Statement of confidentiality

Individual participants' medical or personal information obtained as a result of this study is considered confidential, and disclosure to third parties is prohibited. Participant confidentiality will be further ensured by using identification code numbers to correspond to treatment data in the computer files. Such medical information may be given to the participant's medical team and all appropriate medical personnel responsible for the participant's welfare.

Data generated as a result of this study will be available for inspection on request by the participating physicians, University of Nottingham representatives, REC, local research and development (R&D) departments, and regulatory authorities.

## Data sharing statement

Data originating from the iBEAT-OA trial will be available on request once the results from the trial have been published in a peer-reviewed publication. Researchers interested in accessing the data will need to complete a 'Data Access Proposal Form', and the investigators associated with iBEAT-OA will grant access to the data provided these are used for research purposes only. No personal information of research participants will be shared as part of any data sharing.

## DISCUSSION

Knee OA remains one of the most common forms of OA and affects the majority of the population in the UK. Treatments include non-pharmacological and pharmacological management recommended by NICE. The non-pharmacological management recommends local muscles strengthening, general aerobic fitness, weight loss and using transcutaneous electrical nerve stimulation as an adjunct to other forms of management.[95] The majority of the population requires some guidance as to what exercises they should do and get referred to the Versus Arthritis website for basic exercises. If those exercises fail to make much difference or if patients struggle to understand these exercises, they are referred to a local physiotherapy department in the community. This means that some of these patients will have to travel to local community health centres or hospitals to see a physiotherapist and learn the relevant exercises.

The JA or other similar platform could work in-between the Versus Arthritis leaflet and referral to physiotherapy, thus cutting the cost of travelling, and saving the time of the patient wasted in travelling and the time of physiotherapists so that they can treat other patients with more complex pathologies and needs.

Although iBEAT-OA will not specifically study the economic benefits of delivering the JA intervention instead of face-to-face physiotherapy, it will generate data on its efficacy which will serve as a starting point for any future health economic and cost-effectiveness analysis of such a method for relieving knee OA pain.

The majority of physiotherapists guide their patients on the type of exercises they should follow and review them a few times before the patients are recommended to self-manage OA, and at this stage these patients get discharged. The compliance of patients afterwards can decline as they may stop exercising. There are various reasons for this, including adjusting lifestyle to include these exercises, pain during the exercises, lack of motivation and lack of professional monitoring.[96–98] The JA platform is designed to keep these patients motivated by sending regular emails to remind them and by tracking their progress.

Social media is a powerful platform which offers a connection between users and is a source of social interaction for a range of individuals. This can be used to promote health and to treat patients with OA.[99–101] JA is aimed at using social media such as internet and digital application on a mobile phone to encourage patients with knee OA to self-manage their condition. This programme will educate and train them to stay 'in-control' of their knee OA, which will improve their overall quality of life. This will lead to the overall psychological well-being of our population. This study will also encourage other researchers to study digital health platform, which will be the preferred way of communication and solution to health-related issues for next generations.

This study intends to establish the link between digital exercises, muscles strength, knee inflammation, sleep disturbance, pain and severity of knee OA. The intention is to check if exercising regularly can reduce pain, knee inflammation and sleep disturbance and slow down the progression of knee OA. If this complicated link can be interpreted effectively, we may find a way to reduce the progression of knee OA.

**Author affiliations**
[1]NIHR Nottingham Biomedical Research Centre, School of Medicine, University of Nottingham, Nottingham, UK

2Division of Physiology, Pharmacology and Neuroscience, University of Nottingham School of Medical and Surgical Sciences, Nottingham, UK
3MRC Arthritis Research UK Centre, MSK Ageing Research, Nottingham, United Kingdom
4School of Life Sciences, University of Nottingham, Nottingham, United Kingdom
5NIHR Nottingham Biomedical Research Centre, School of Medicine, University of Nottingham, Nottingham, United Kingdom
6Academic Rheumatology, University of Nottingham, Nottingham, United Kingdom

**Acknowledgements** We would like to thank Dr Michelle Hall (Assistant Professor, Division of Physiotherapy and Rehabilitation Sciences, University of Nottingham) and Tony Kelly (Research Metrologist, Academic Rheumatology, University of Nottingham) for their valuable contribution to designing, monitoring and implementing this study.

**Contributors** SG is the primary author and all other authors are secondary. AMV is the main supervisor and is leading this project. PG and AA are secondary supervisors. All authors have equally contributed to this article.

**Funding** This study is part-funded by the Pain Centre Versus UK (University of Nottingham) [Grant numbers 21960, 18769] and by Nottingham Biomedical Research Centre (BRC).

**Competing interests** None declared.

**Patient consent for publication** Not required.

**Ethics approval** The study has received approval from the Research Ethics Committee (REC) (ref: 18/EM/0154), Health Research Authority (HRA) (protocol no: 18021) and the Nottingham University Hospitals NHS Trust Research & Innovation (R&I) department (ref: 18RH004). Any modification to the approved protocol will require resubmission of modifications and further approval from the REC and the sponsor.

**Provenance and peer review** Not commissioned; externally peer reviewed.

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
