## [Reviewer comments · BMJ Open]

ARTICLE DETAILS

TITLE (PROVISIONAL)	A study protocol for a randomised controlled trial evaluating the efficacy of an Internet-Based Exercise programme Aimed at Treating Knee Osteoarthritis (iBEAT-OA) in the community
AUTHORS	Gohir, Sameer; Greenhaff, Paul; Abhishek, A; Valdes, A

VERSION 1 – REVIEW

REVIEWER	Philippa Nicolson Centre for Rehabilitation Research in Oxford Nuffield Department of Orthopaedics, Rheumatology and Musculoskeletal Sciences University of Oxford United Kingdom
REVIEW RETURNED	30-Apr-2019

GENERAL COMMENTS	Thank you for the opportunity to review this study protocol for a randomised controlled trial evaluating the efficacy of an internet based exercise program for knee osteoarthritis. This protocol outlines an ongoing trial, however, a number of major aspects of the article require further clarification and explanation. Reporting guidelines are not referenced – the SPIRIT checklist for protocol reporting (http://www.equator-network.org/reporting-guidelines/spirit-2013-statement-defining-standard-protocol-items-for-clinical-trials/), and the TIDieR checklist (http://www.equator-network.org/reporting-guidelines/tidier/) for intervention reporting should be utilised. Throughout the manuscript reference is made to BOA, The OA School, Joint Academy and iBEAT-OA – however the link between all of these / how the iBEAT-OA intervention differs / why it differs is not clear. Introduction • A stronger case needs to be made to justify the need for this study / the novelty of this study. No reference is made to two recent large RCTs examining internet delivered exercise interventions for knee osteoarthritis – it is important to consider their findings (Allen KD et al. Physical therapy vs internet-based exercise training for patients with knee osteoarthritis: results of a randomized controlled trial. Osteoarthritis Cartilage. 2018;26(3):383-96.) (Bennell KL et al. Effectiveness of an Internet-Delivered Exercise and Pain-Coping Skills Training Intervention for Persons With Chronic Knee Pain: A Randomized Trial. Annals of Internal Medicine. 2017;166(7):453-62.)
--

	 • Throughout the manuscript 'arthritis' and 'osteoarthritis' are both used – consistency with using osteoarthritis or OA is needed. Sample size and justification  • Justification is needed for the use of 75% statistical power rather than the conventional minimum of 80%. • No alpha is provided, precluding replication of sample size calculation. Outcomes  • The primary outcome is stated as 'pain perception in knee OA' – however it is not clear which outcome measure is to be used to assess this? • 'NRS' is mentioned in the Description of the Intervention but not in the list of outcome measures. Is this NRS average pain / worst pain / pain with walking? • Justification for the included outcomes is needed in relation to the existing core outcome domains and recommended performance tests: (Smith TO et al. The OMERACT-OARSI Core Domain Set for Measurement in Clinical Trials of Hip and/or Knee Osteoarthritis. J Rheumatol. 2019.) (Dobson F et al. OARSI recommended performance-based tests to assess physical function in people diagnosed with hip or knee osteoarthritis. Osteoarthritis Cartilage. 2013;21(8):1042-52.) Start and end dates Please supply months rather than seasons for international readers. Figure 1  • Research question – needs to be more clearly defined • Outcome measures – NRS – define; sleep – how assessed? • Time period – need to justify why a 6 week intervention is appropriate as very short compared to other exercise interventions for knee OA. Figure 3 Inclusion and exclusion criteria  • ? minimum pain level for clinical diagnosis of knee osteoarthritis • A radiographic diagnosis of OA is stated, however justification should be made for using a KL score of ≥ 1 as this is not 'established knee OA'. • Exclusion criteria – mental illness – how is this assessed? Does this include depression as this is common with knee OA?
--	---

REVIEWER	Christian Barton La Trobe Sport and Exercise Medicine Research Centre
REVIEW RETURNED	17-May-2019

GENERAL COMMENTS	Thank you for the opportunity to review this protocol related to an exciting study in the field of OA. I have a number of comments for consideration before accepting this for publication. Most importantly, this study is not reproducible based on the Methods described related to the intervention and outcome measures. A protocol paper should allow reproduction of the study. General comments:  1. Sentence structure requires a thorough check throughout – perhaps done at an editorial level, or by the authors prior to resubmission.
--

2. Abstract – reference is made to muscle strengthening exercises. Most studies do not provide adequate prescription (load, duration, etc.) to induce strength gains. At 6-weeks duration, I struggle to see this study providing a legitimate strength program. That is Ok since it is unclear what the key ingredient in exercise might be, just needs to be clear and true to exercise science principles in our reporting.
3. Interventional – this term is used throughout, perhaps intervention is the better term

STRENGTH AND LIMITATION OF THIS STUDY

4. Patellofemoral syndrome is not a recommended term. Suggest replacing with patellofemoral pain
5. Final sentence – delete one of the “only” words

INTRODUCTION

6. Better details of evidence for exercise-therapy should be provided to justify this study. What are the effect sizes etc. compared to no intervention
7. Second paragraph – please define what ineffective or unsafe exercise is? Are the better supporting references here?
8. Third paragraph – opening sentence. I am unsure of the point being made here.
9. Third paragraph – points in this paragraph are poorly references in general – does this relate to NICE/UK guidelines – I think this could be more clearly articulated
10. “maximise efficacy testing” what do you mean here?
11. Cost-effectiveness is mentions, yet I do not see any plans for evaluation here.
12. Third paragraph – final sentence. 94% of who???
13. What are the actual changes in pain with SOASP? How is this linked to the program being tested here? i.e. Joint Academy
14. It might be worth mentioning other implementation initiatives – GLA:D, ESCAPE – including their limitations (i.e. not available online, possibly resource intensive).
15. Justification for the need of an XRay for inclusion is interesting, considering NICE guidelines in the UK. Is there any evidence to suggest that outcomes would differ with or without XRay changes? Why would exercise programs from other studies mentioned risk irritating symptoms of people with knee OA. I think justification for the study being conducted here based on this points is shaky.
16. Great work re inclusion of general health variables – well done!

SAMPLE SIZE AND JUSTIFICATION

17. 75% power is low compared to commonly accepted 80% - can this be justified?
18. 12/10 may be the average difference, but this difference may not be the best figure to sue for sample size calculation. What is a minimal important difference (e.g. determined by PPI, or from other literature)?

INTERVENTION

19. “The efficacy is reported elsewhere” – please provide more details in this paper
20. Second paragraph – “closed” not “close”
21. Generally this is poorly described which is a real problem in a protocol paper. The study, including intervention should be replicable from a protocol paper. Otherwise what is the purpose of publishing it. As a guide, use recent review by Bartholdy et al (2019 - <https://www.sciencedirect.com/science/article/pii/S1063458418314705>) as a guide on what should be reported. An appendix with pictures

	and/or links to videos would be helpful. Similarly for education components, please provide more details of what these provide. 22. This is not likely changeable now, but I question the 6-week follow up to determine 'efficacy' for an intervention in a chronic disease. This should be discussed as a limitation OUTCOME MEASURES 23. More details needed here to allow replication – for example, for QST, which body sites and how? How is US assessment completed? Which biomarkers? Perhaps online sup files are needed to help here?
--	---

VERSION 1 – AUTHOR RESPONSE

Reviewer: 1

Reviewer Name

Philippa Nicolson

Institution and Country

Centre for Rehabilitation Research in Oxford

Nuffield Department of Orthopaedics, Rheumatology and Musculoskeletal Sciences

University of Oxford

United Kingdom

Please state any competing interests or state 'None declared':

None declared

Please leave your comments for the authors below

Thank you for the opportunity to review this study protocol for a randomised controlled trial evaluating the efficacy of an internet based exercise program for knee osteoarthritis.

This protocol outlines an ongoing trial, however, a number of major aspects of the article require further clarification and explanation.

Reporting guidelines are not referenced – the SPIRIT checklist for protocol reporting (<http://www.equator-network.org/reporting-guidelines/spirit-2013-statement-defining-standard-protocol-items-for-clinical-trials/>), and the TIDieR checklist (<http://www.equator-network.org/reporting-guidelines/tidier/>) for intervention reporting should be utilised.

Authors Response and Action: We thank the reviewer for pointing this out. We have submitted the checklist as suggested.

1- Throughout the manuscript reference is made to BOA, The OA School, Joint Academy and iBEAT-OA – however the link between all of these / how the iBEAT-OA intervention differs / why it differs is not clear.

Authors' Response and Action: We thank Dr Nicolson for her valuable feedback and for pointing out this. We agree that the use these names can create some confusion. We have added text explaining that Joint Academy is the name of the digital health App that delivers BOA. BOA is the face-to-face exercise intervention developed in Sweden and iBEAT-OA is the first RCT using Joint Academy. The link between iBEAT-OA and BOA has been elaborated in more detail under section introduction and intervention now and hopefully will be clearer to the reader.

Introduction

2-• A stronger case needs to be made to justify the need for this study / the novelty of this study. No reference is made to two recent large RCTs examining internet delivered exercise interventions for knee osteoarthritis – it is important to consider their findings (Allen KD et al. Physical therapy vs internet-based exercise training for patients with knee osteoarthritis: results of a randomized controlled trial. *Osteoarthritis Cartilage*. 2018;26(3):383-96.)

(Bennell KL et al. Effectiveness of an Internet-Delivered Exercise and Pain-Coping Skills Training Intervention for Persons With Chronic Knee Pain: A Randomized Trial. *Annals of Internal Medicine*. 2017;166(7):453-62.)

Authors' Response and Action: We thank the reviewer for highlighting this. We have now included reference to both studies. Please note findings from the first study (a non inferiority trial affected with OA) has been mentioned in the updated version of manuscript. The second study concerned knee pain which may have included participants with non-OA (such as soft tissues injuries), thus limiting the relevance of the findings to this study. We have highlighted that none of the previous studies has investigated the link between pain sensitivity/ central sensitization and exercise. Other non-OA forms of knee pain are unlikely to have the same pattern of chronification and central sensitization and hence the need for the current study design.

With regards to the justification for this study, we have included an additional paragraph in the Introduction emphasizing the novelty of this study.

3-• Throughout the manuscript 'arthritis' and 'osteoarthritis' are both used – consistency with using osteoarthritis or OA is needed.

Authors' Response and Action: Thank you. This has been corrected and we now refer to osteoarthritis for consistency.

Sample size and justification

4-• Justification is needed for the use of 75% statistical power rather than the conventional minimum of 80%.

- No alpha is provided, precluding replication of sample size calculation.

Authors' Response: We apologise for the confusing language used to describe the statistical power.

To be clear, the sample size required to achieve 80% power with an alpha of 0.05 is 132 (corresponding to 0.49 SD). This conclusion is based on a systematic review of different type of different types of exercise intervention for OA [Fransen M, McConnell S, Harmer AR, Van der Esch M, Simic M, Bennell KL. Exercise for osteoarthritis of the knee: a Cochrane systematic review. *British Journal of Sports Medicine*. 2015;49(24):1554-7.]. Therefore assuming a 12% drop-out rate, this require we recruit 150 participants.

However, a Swedish study using specifically Joint Academy and NRS [Nero H, Dahlberg J, Dahlberg EL. A 6-Week Web-Based Osteoarthritis Treatment Program: Observational Quasi-Experimental Study. *J Med Internet Res*. 2017;19(12):e422] reported an effect size of 1.3 on the NRS which corresponds to 0.56 SD (average SD 2.3). Our study with an expected 60 participants per arm at the end of the 6-week intervention has 86% power to achieve such an effect size with an alpha of 0.05.

Authors' Action: Sample size section has been updated with the references mentioned above to justify the sample size.

Outcomes

5-• The primary outcome is stated as 'pain perception in knee OA' – however it is not clear which outcome measure is to be used to assess this?

Authors' Response and Action: This had been changed to reflect the use of the Numerical Rating Scale (NRS).

6-• 'NRS' is mentioned in the Description of the Intervention but not in the list of outcome measures. Is this NRS average pain / worst pain / pain with walking? –

Authors' Response and Action: We have now clarified that the NRS refers to average pain on the day of assessment and it has been added to the list of outcome measures.

7-• Justification for the included outcomes is needed in relation to the existing core outcome domains and recommended performance tests:

(Smith TO et al. The OMERACT-OARSI Core Domain Set for Measurement in Clinical Trials of Hip and/or Knee Osteoarthritis. J Rheumatol. 2019.)

(Dobson F et al. OARSI recommended performance-based tests to assess physical function in people diagnosed with hip or knee osteoarthritis. Osteoarthritis Cartilage. 2013;21(8):1042-52.)

Authors Response and Action: We thank the reviewer for pointing this out. These references have been added under Research assessment section.

Start and end dates

8-Please supply months rather than seasons for international readers.

Authors Response and Action: This has been changed as requested.

9-Figure 1

- Research question – needs to be more clearly defined
- Outcome measures – NRS – define; sleep – how assessed?
- Time period – need to justify why a 6 week intervention is appropriate as very short compared to other exercise interventions for knee OA.

Authors' Response and Action: Many thanks for these helpful comments. These points were addressed earlier in the main article. Additionally, the terms are defined as a footnote to the figure and highlighted with an asterisk. The rationale for the 6-week intervention has now also been explained under the section 'follow up duration'.

10-Figure 3 Inclusion and exclusion criteria

- ? minimum pain level for clinical diagnosis of knee osteoarthritis
- A radiographic diagnosis of OA is stated, however justification should be made for using a KL score of ≥ 1 as this is not 'established knee OA'.
- Exclusion criteria – mental illness – how is this assessed? Does this include depression as this is common with knee OA?

Authors' Response and Action: Many thanks Re "Minimum pain level for clinical diagnosis of osteoarthritis". We are recruiting the participants on the basis of their GPs having diagnosed the participant with knee OA or based on x-rays findings and not on the minimum pain level. Therefore there are no criteria limiting the minimum pain level for diagnosing OA or recruiting participants into the study.

The radiographic diagnosis is based on minimum changes on X-rays. The article [PS Emrani et al. Joint space narrowing and Kellgren–Lawrence progression in knee osteoarthritis: an analytic literature synthesis. Osteoarthritis and Cartilage 2008] has included studies which assessed knee OA K–L ≥ 1 and K–L ≥ 2 and their progression.

While K/L K–L ≥ 1 can be described as 'not established OA' it has been shown to be strongly predictive of the development of OA and is indicative of radiographic signs of OA (Kerkhof et al. Prediction model for knee osteoarthritis incidence including clinical, genetic and biochemical risk factors. BMJ. 2014;73(12):2116-21).

Lastly, mental illness (excluding depression) is established based on participant past medical history. Furthermore, participants who have received from their GP, or from a specialist, a diagnosis of Schizophrenia or multiple personality disorders will be excluded from the study. This information has been added to Figure 3.

Reviewer: 2

Reviewer Name

Christian Barton

Institution and Country

La Trobe Sport and Exercise Medicine Research Centre

Please state any competing interests or state 'None declared':

None declared

Please leave your comments for the authors below

Thank you for the opportunity to review this protocol related to an exciting study in the field of OA. I have a number of comments for consideration before accepting this for publication. Most importantly, this study is not reproducible based on the Methods described related to the intervention and outcome measures. A protocol paper should allow reproduction of the study.

Authors' Response and Action:

The main reason to keep the protocol article concise was to comply with the word count given by Open BMJ. However, as we have already exceeded that limited, therefore, we will try and provide information so that it can help with the reproduction of the study.

General comments:

1. Sentence structure requires a thorough check throughout – perhaps done at an editorial level, or by the authors prior to resubmission.

Authors' Response and Action:

Corresponding Author and other authors have checked this.

2. Abstract – reference is made to muscle strengthening exercises. Most studies do not provide adequate prescription (load, duration, etc.) to induce strength gains. At 6-weeks duration, I struggle to see this study providing a legitimate strength program. That is Ok since it is unclear what the key ingredient in exercise might be, just needs to be clear and true to exercise science principles in our reporting.

Authors' Response and Action: Many thanks. The rationale for 6 weeks intervention has been explained now under the section Follow up duration in the Methods and Analysis.

3. Interventional – this term is used throughout, perhaps intervention is the better term

Authors' Response and Action: we have changed interventional to intervention.

STRENGTH AND LIMITATION OF THIS STUDY

4. Patellofemoral syndrome is not a recommended term. Suggest replacing with patellofemoral pain

Authors' Response and Action: An editorial request was to use bullet points and short sentences and therefore this term has been removed from the manuscript.

5. Final sentence – delete one of the “only” words

Authors' Response and Action: This has been edited as requested

INTRODUCTION

6. Better details of evidence for exercise-therapy should be provided to justify this study. What are the effect sizes etc. compared to no intervention

Authors' Response and Action: Further information have been added to the Introduction as suggested.

7. Second paragraph – please define what ineffective or unsafe exercise is? Are the better supporting references here?

Authors' Response and Action: Additional references have been added. There is limited evidence on 'ineffective, unsafe or harmful knee exercises'.

8. Third paragraph – opening sentence. I am unsure of the point being made here.

Authors' Response and Action: We apologise for the lack of clarity. The point being made is that exercises for knee pain may not be specific to knee OA. We have reworded the sentence accordingly.

9. Third paragraph – points in this paragraph are poorly references in general – does this relate to NICE/UK guidelines – I think this could be more clearly articulated

Authors' Response and Action: Thank you for this valuable comment. We are trying to explain what is offered as 'standard treatment' under NHS care and what care is it lacking. The lack of specificity of exercise interventions for knee OA highlights the importance of finding an exercise regimen suitable for knee OA.

10. "maximise efficacy testing" what do you mean here?

Authors' Response and Action: Testing word has been removed

11. Cost-effectiveness is mentions, yet I do not see any plans for evaluation here.

We mentioned cost-effectiveness because a digital health care intervention does not involve person to person interactions that the National Health Service would have to pay for to deliver physiotherapy intervention. The evidence on efficacy of pain relief due to digital health care intervention could then be used by other researchers to develop health economic arguments regarding the value of this digital health intervention for OA in the specific UK setting. We have clarified this in the text.

12. Third paragraph – final sentence. 94% of who???

Authors' Response and Action: Thanks for pointing out and it has been explained as "This intervention has been rated good to very good by 94% of patients at reducing the pain related to knee OA and has an excellent safety profile and acceptability".

13. What are the actual changes in pain with SOASP? How is this linked to the program being tested here? i.e. Joint Academy

Authors' Response and Action: The actual change in pain has been mentioned in the last line of third paragraph of Introduction and many thanks for the prompt.

iBEAT-OA and link with SOASP has been explained in third paragraph of Introduction, 'The exercise intervention (iBEAT-OA), which we will use in this project, is part of a web-based program derived from the Supported Osteoarthritis Self-Management Programme (SOASP).' We have made some changes to make it clear to our readers.

14. It might be worth mentioning other implementation initiatives – GLA:D, ESCAPE – including their limitations (i.e. not available online, possibly resource intensive)

Authors' Response and Action: We thank the reviewer for pointing this out. We have now added text to the manuscript regarding other initiatives. We have also highlighted that none were developed to be delivered via a web or smartphone App and hence are not appropriate for the current study aims. Following text has been added to the manuscript;

"There exist a number of other exercise programmes within care programmes for people with knee OA pain. Among these are the:

Good Life with OA in Denmark (GLA:D) programme, which is a registry based study that implemented clinical guidelines (patient education and exercise but not weight loss) to knee OA patients (37). The

programme consists of a 2-day training course for physiotherapists (PTs), including training to diagnose and deliver OA care and an 8-week supervised exercise intervention for OA patients, with a minimum intervention of 3 sessions in total. However, GLA:D does not deliver intervention via a web or smartphone App.

The ESCAPE (Enabling Self-management and Coping with Arthritic Pain using Exercise) app is a support tool for people who have already attended a person-to-person ESCAPE programme (20) and enables participants to continue to exercise safely in the home environment following the person-to-person programme. A limitation of this approach is that the app is a support tool to the main programme and provided for reminding exercise. It cannot be used as the sole basis for treatment and care of knee OA. Their recommendations are to use it in conjunction with the advice and professional judgment of GP or other healthcare practitioner, which is the limitation of this App."

15. Justification for the need of an XRay for inclusion is interesting, considering NICE guidelines in the UK. Is there any evidence to suggest that outcomes would differ with or without XRay changes? Why would exercise programs from other studies mentioned risk irritating symptoms of people with knee OA. I think justification for the study being conducted here based on this points is shaky. Authors' Response and Action: We agree with the Reviewer, however, knee pain can originate from many other sources (e.g. patellofemoral pain, ITB syndrome etc). Although NICE guidelines do not specifically address X-rays, our study is also collecting a series of inflammatory markers and sleep information. By selecting individuals with radiographic changes, we hope to exclude other forms of pathology that result in knee pain, which could confound the findings. Although this does not preclude that the intervention may be effective also in those knee pain conditions.

16. Great work re inclusion of general health variables – well done!
Authors' Response and Action: Many thanks.

SAMPLE SIZE AND JUSTIFICATION

17. 75% power is low compared to commonly accepted 80% - can this be justified?
We apologise for the lack of clarity, which we have corrected. Please see the reply to Reviewer # 1 point #4, who raised the same point.

18. 12/10 may be the average difference, but this difference may not be the best figure to sue for sample size calculation. What is a minimal important difference (e.g. determined by PPI, or from other literature)?

Authors' Response and Action: Salafi et al (2004) [Salaffi et al, Minimal clinically important changes in chronic musculoskeletal pain intensity measured on a numerical rating scale. Eur J Pain. 2004 Aug;8(4):283-91.] have reported that a difference of -2 in the NRS corresponds to a clinically important improvement ("much better") in musculoskeletal pain. Given a standard deviation of 2.3 for NRS in the previous Joint Academy study [Nero H, Dahlberg J, Dahlberg EL. A 6-Week Web-Based Osteoarthritis Treatment Program: Observational Quasi-Experimental Study. J Med Internet Res. 2017;19(12):e422.] a difference of -2 in the NRS corresponds to 0.87 standard deviations, our study with 60 participants in the control arm and 60 in the treatment arm has therefore >99% power to detect such a difference at an alpha level of 0.05.

INTERVENTION

19. "The efficacy is reported elsewhere" – please provide more details in this paper

Authors' Response and Action: we have added following to the manuscript:

"The compliance of this self-management programme has been reported as good. 62% of 20,200 patients reported daily use of this programme at three months follow-up however, this percentage drops to 37% at 12 months review."

20. Second paragraph – “closed” not “close”

Authors' Response and Action: Corrected.

21. Generally this is poorly described which is a real problem in a protocol paper. The study, including intervention should be replicable from a protocol paper. Otherwise what is the purpose of publishing it. As a guide, use recent review by Bartholdy et al (2019 - <https://www.sciencedirect.com/science/article/pii/S1063458418314705>) as a guide on what should be reported. An appendix with pictures and/or links to videos would be helpful. Similarly for education components, please provide more details of what these provide.

Authors' Response and Action: Many thanks for your valuable comment. It is a fair point so we have added a link to Joint Academy where the reader can achieve better appreciation of the digital intervention, and to the BOA site where more detailed information regarding the nature of the intervention can be obtained.

22. This is not likely changeable now, but I question the 6-week follow up to determine 'efficacy' for an intervention in a chronic disease. This should be discussed as a limitation

Authors' Response and Action: Many thanks. This is a fair and valid point. Six weeks intervention was selected based upon a quasi-study cited in this manuscript (third paragraph of the Introduction). As per your suggestion, we have added this in the section focused on study limitation.

OUTCOME MEASURES

23. More details needed here to allow replication – for example, for QST, which body sites and how? How is US assessment completed? Which biomarkers? Perhaps online sup files are needed to help here?

Authors' Response and Action: This has been added as supplementary information with the document title 'Assessment explained'.

VERSION 2 – REVIEW

REVIEWER	Philippa Nicolson Nuffield Department of Orthopaedics, Rheumatology and Musculoskeletal Sciences, University of Oxford, UK
REVIEW RETURNED	26-Jul-2019
GENERAL COMMENTS	The authors have adequately and clearly addressed my previous comments, thank you very much. Prior to publication I recommend the article undergoes editing of sentence structure and grammar to improve readability.